# OpenReview forum: "Adaptive LLM Safety via Inference-Time System-Level Optimization"
_TMLR — Rejected by TMLR_

### Review · Reviewer_nZ2s · 2026-03-12

**Summary Of Contributions:**

the paper proposes and studies modular defense approaches for LLMs that can be applied at inference time and adapted to the info one has then about the threat. the threats studied include jailbreaks, i.e., prompts that bypass the original alignment of the model, and sensitive information disclosure. the approach consists of multiple modules around a core LLM.
the approach is described on a high level, and then experiments are conducted, evaluating the approach on various existing benchmarks, baselines and metrics, but also a small ablation study. based on this, the evidence for the proposed approach is mostly empircal/experimental.

the strength of the paper is that it addresses an interesting problem - LLM safety in general - but also the aspect of inference-time adaptability based on few-shot approaches in particular. the approach of combining modules intuitively makes sense, since modular approaches seem more adaptable and possibly interpretable. in terms of writing, a threat model is clearly introduced which helps the problem formulation and the intuition. extensive experiments are conducted.

**Audience:**

Yes

**Audience Explanation:**

LLM safety is an unsolved, relevant problem.

**Broader Impact Concerns:**

-

**Claims And Evidence:**

No

**Claims Explanation:**

overall, it's not clear to me what to make of this paper. it does contain some interesting thoughts, and quite some experimental evaluation. but to me it's really unclear what **insights** can be gained from this. i could well imagine that with just minor tweaks, like using another backbone here or there, or using a different benchmark, or a different subset on some benchmark (see also the comment on the inference-time adaptability below), results could already look completely differently. in addition to that, for me, the style of writing makes it even more unclear, as detailed below. from my personal view, in its current form, this could be a technical report containing experimental numbers of some method, whereas i'm not sure this is enough for a scientific paper. but again, this is just my personal view and i'm aware that there is a general trend towards such papers in the ML community.

specifically on the claims and their match to the evidence, as well as clarity:

on the first claim:
- it is claimed "to empirically demonstrate that an adaptive, system-level approach to security achieves a superior safety-utility trade-off compared to static, model-level defense". first, on the formulation side, i feel it is impossible to empirically, based on a few benchmarks demonstrate such a broad claim. maybe the term "case study" already relativizes it a bit, but i recommend to be more modest and clear in general. i don't want to prescribe a specific formulation, but the claim needs to be more limited to the scope of the benchmarks that are actually studied. without theory and clear preconditions, it's not possible to derive such a general claim from a limited benchmark study.
- i feel that there is too much unclarity and uncertainty about the experimental setup and conclusions to say that this claim is well-supported by evidence. as a reader, in the experiment section, i feel a bit overwhelmed by abbreviations and numbers and it's really hard to *judge how significant* these numbers are, and what the underlying, generalizable *insights* are, if almost nothing is explained: what specifically are the benchmarks about? what specifically are the metrics about, what do they measure? what are limitations of benchmarks/metrics on such complex evaluations tasks? etc. maybe readers super-familiar to this topic directly have a better hunch on all this, but i got somewhat lost.
- one specific problem is about the baselines performance: where are the metrics values of the baselines taken from? if the benchmarking is such a central part of the paper, then these things should be made really clear. are the numbers taken from the original papers? or based on own experiments with the existing methods? **which backbones are used in these baselines?** maybe differences between the proposed method and the baselines come from the baselines' backbones simply being older, with less training data and less capacity? this is absolutely crucial to asses the power of the proposed methodological ideas, to see if these ideas are the helpful factor, or if it's just a better backbone. i see one half sentence on this in Sec5: "Baselines ... using pre-trained models from Che et al. (2025)" - but what does this mean specifically? Is this the same backbone as the present method uses? Are the experiments conducted by the authors, or just numbers copied over? And for the jailbraking, nothing seems to be said about this aspect.
- the claim of (inference-time) adaptability is central to the paper, and it is an interesting direction. however, it feels like this claim is only supported by sec5.3, if i see correctly, and this section is really short. it does point to fig6, which contains few numbers that does go in the direction of supporting the claim, but if i understand correctly, the evaluation set contains just the "15 strongest attacks" of the benchmark, so just 15 samples, and all from one and the same benchmark. i think this is far too little to support a substantial claim here.
- the ablation study goes in the right direction, but some ablations like removing the elevator are actually still rather close to the un-ablated system. it's true that it's just close in absolute percentage points, while it is more significant when looking at it relatively. however, i think statistical significance is often to be measured in absolute numbers (additionally, low percentages may mean low numbers of samples meaning low confidence).

on the second claim:
- this again is a claim that sounds really general, but to me it feels like what is actually done is to apply an existing, widely used, off-the-shelve prompt-optimization to a specific problem an a few datasets. this can be of some limited value, but this limitedness needs to be clearly reflected in the claim.

generally, i find the "writing allocation" to be imbalanced: the paper spends quite some time on parts that are rather straight-forward or just adaptations of well-existing methods (in particular sec4.2), and then on the central parts of experimental setups, benchmarks, metrics, baselines, and what the scope and limitations of each of them are, the reader gets very little clue.

**Requested Changes:**

here are some things that might be improved, but i can't really make specific requests, as it really depends on in which direction the paper is supposed to go, which is not clear to me:
- generally, to address a somewhat broad audience, for readers like me, who are not fully into the matter of LLM-safety, some terms and formulations need to be made more concrete, or definitions/examples could be given. generally, i recommend to give definitions or at least short preliminary explanations for technical terms ideally right at their first occurence.
- when i read the introduction without knowing the subsequent sections: "exploites ranging from prompt injection and jailbreaking to sensitive data exfiltration" - i have no idea what "jailbreaking" means here specifically (though i did hear the term before). for the other terms i have a vague imagination, but even for them a half-sentence explanation might be sensible. "jailbreaking" is defined later on but i strongly recommend to introduce technical terms at least briefly right when they are used first, otherwise the reader gets lost rather easily.
- the sentence "Meanwhile, the concept of model scaling (Kaplan et al., 2020; Ke et al., 2025) has been central to enhancing LLM capabilities, with strategies categorized across three key dimensions:" feels a bit vague to me. one has a rough idea what "model scaling" could mean, but i would wish for more concrete statements.
- "... progress in security has remained disproportionately confined to training-time." what does this mean? does this mean "so far, to improve security aspects of LLMs, many approaches focused on training the models in a specific way to this end ..."?
- the content of the benchmarks is quite important to understand what this is all about. unfortunately, the benchmarks are described quite briefly. e.g., what does the abbreviation "WMDP" stand for? i see there is some more info in appendix C, but this is sparse as well and i'd wish for a concise explanation of that very basics of the experimental setup in the main paper. i also can look at the referred paper for the benchmark, and then things become more clear, but i'd strongly suggest to put such important information right there for the reader.
- what is the "compliance" metric in p7-9 / tab5? again a term that somehow is central to the argument on these pages but it's nowhere introduced.
- please introduce all abbreviations at least once explicitly, in particulary "PII" on p4.

---

> ### Author Response · Authors · 2026-04-08
> **Rebuttal to Reviewer nZ2s: Strengthening Validity and Scope Through Expanded Analysis**
>
> We thank the reviewer for the thorough and thoughtful reading of our paper. We appreciate the candid feedback and agree that several aspects of the presentation can be substantially improved. Many of the reviewer's concerns point to the same root cause: the paper spends too much space on methodology that is largely borrowed from existing frameworks, and too little space explaining the experimental setup, what the metrics measure, and what the results actually tell us. We will address each concern below and describe concrete changes we will make to the revised paper.
>
> ## C1: Overbroad Claims
> > it is claimed "to empirically demonstrate that an adaptive, system-level approach to security achieves a superior safety-utility trade-off..."... i recommend to be more modest and clear in general. i don't want to prescribe a specific formulation, but the claim needs to be more limited to the scope of the benchmarks that are actually studied.
>
> We appreciate the reviewer's concern about overbroad claims, but respectfully argue that the appropriate response is careful scoping rather than radical conservatism. In ML research, empirical claims that are well-supported across multiple model families, threat types, and benchmarks carry scientific value even without theoretical guarantees. This is the standard form of evidence in the field.
>
> Our study evaluates three model families spanning a 9x range in parameter count (LLaMA-3-8B, Qwen2.5-72B, DeepSeek-R1 8B and 70B), two fundamentally different threat types (jailbreaking and sensitive information disclosure), and four benchmarks including one zero-shot transfer evaluation on a held-out distribution (Wildjailbreak). The consistency of our findings across these diverse settings is itself scientific evidence for generalizability, beyond what any single benchmark or model could provide.
>
> We agree, however, that the current language in some places implies universality that our evidence does not support. We will revise claims in two ways. First, we will replace language that implies universal laws ("adaptive systems achieve superior tradeoffs") with language that makes the empirical scope explicit ("across the models and benchmarks we study, we find that..."). Second, we will add a paragraph framing our contribution as an existence proof: we demonstrate that inference-time system-level optimization is a viable and effective approach to LLM safety, that it generalizes across model families and threat types, and that it enables rapid post-deployment adaptation with very few examples. These are meaningful scientific findings that inform the community about a class of approaches worth pursuing, without overclaiming universality.
>
> We believe this framing is both scientifically honest and appropriately strong. The evidence we present is among the most comprehensive evaluations of inference-time defense systems in the literature, and understating its implications would do a disservice to the contribution.

---

> > ### Author Response · Authors · 2026-04-08
> > **Rebuttal to Reviewer nZ2s: Strengthening Validity and Scope Through Expanded Analysis (cont'd)**
> >
> > ## C2: Experimental Setup Clarity: Baselines, Backbones, and Metric Definitions
> > > as a reader, in the experiment section, i feel a bit overwhelmed by abbreviations and numbers...  what specifically are the benchmarks about? what is the "compliance" metric...? please introduce all abbreviations...
> >
> > We thank the reviewer for identifying these gaps, which we agree make it difficult to assess the validity of the comparisons. We will make the following clarifications in the revised paper:
> >
> > **Baselines and backbones**: For the unlearning baselines (GradDiff, RMU, RMU-LAT, ELM, TAR), the pre-trained unlearned model checkpoints are obtained directly from Che et al. (2025), who provide standardized unlearned versions of LLaMA-3-8B across methods. AegisLLM uses the same LLaMA-3-8B as its Responder, making the backbone identical across all compared methods. We did not re-run the baseline methods ourselves; we used the published checkpoints. For the jailbreaking baselines, we ran all evaluations ourselves using the same LLaMA-3-8B base model where applicable. We will add a dedicated paragraph in Section 5 and a footnote in Table 2 clarifying this, and will add a column to Table 2 indicating which backbone each method uses.
> >
> > **Metric definitions**: We will add brief explanations of each metric at its first appearance in the main text. Specifically: WMDP accuracy is reported where lower is better (closer to 25% random chance indicates better suppression of hazardous knowledge); StrongREJECT score measures the quality of harmful content elicited by jailbreak attacks on a 0-1 scale where lower is better; PHTest compliance measures the fraction of pseudo-harmful but actually benign queries that the system answers correctly rather than refusing, where higher is better. The "compliance" metric in Tables 4 and 5 refers to full compliance on PHTest. We will define this explicitly at its first use rather than leaving it implicit.
> >
> > **Benchmark descriptions**: We will expand the benchmark descriptions in Section 5 to include what each benchmark tests, what the evaluation protocol is, and what the key limitations of each benchmark are. Currently this information is spread across the appendix and the main text in an uneven way.
> >
> > We provide a list of the abbreviations used in the paper as follows:
> > *   **LLM:** Large Language Model
> > *   **PII:** Personally Identifiable Information
> > *   **WMDP:** Weapons of Mass Destruction Proxy (benchmark)
> > *   **TOFU:** A Task of Fictitious Unlearning (benchmark)
> > *   **MMLU:** Massive Multitask Language Understanding (benchmark)
> > *   **MT-Bench:** Multi-Turn Benchmark (for conversation quality)
> > *   **PHTest:** Pseudo-Harmful Test (benchmark)
> > *   **ASR:** Attack Success Rate
> > *   **FN:** False Negative
> > *   **FP:** False Positive
> > *   **RLHF:** Reinforcement Learning from Human Feedback
> > *   **API:** Application Programming Interface
> > *   **TPE:** Tree-structured Parzen Estimator
> > *   **CoT:** Chain of Thought
> >
> >
> > ## C3: On the Evidential Support for the Adaptability Claim
> > > the claim of (inference-time) adaptability is central to the paper... however, it feels like this claim is only supported by sec5.3... the evaluation set contains just the "15 strongest attacks"...  i think this is far too little to support a substantial claim here.
> >
> >
> > We agree that Section 5.3 is short and that Figure 6 alone (covering 15 attacks from one benchmark) provides limited support for the adaptability claim. We will expand this section substantially in the revised paper by explicitly connecting two results that together operate at different ends of the sample efficiency spectrum. First, Figure 6 shows few-shot adaptation: with 5-15 labeled examples of a new attack type, the system meaningfully improves its defenses against that attack while maintaining low false refusal rates. Second, Table 5 shows zero-shot transfer: prompts optimized on StrongREJECT and PHTest using 50 total examples transfer directly to Wildjailbreak (a benchmark with over 5,700 unique attack clusters from a different distribution) without any additional optimization. We will reframe Section 5.3 to present these as complementary evidence for adaptability, and will clarify the scope of each result explicitly rather than leaving the reader to connect them.
> > We also note the reviewer's concern about the evaluation set size in Figure 6. The 15 attacks evaluated are specifically the strongest attacks in StrongREJECT. They are selected because they are the most resistant to the base optimized policy. Improvement on this subset is therefore a conservative lower bound on overall adaptation effectiveness. We will add this clarification to the figure caption.

---

> > > ### Author Response · Authors · 2026-04-08
> > > **Rebuttal to Reviewer nZ2s: Strengthening Validity and Scope Through Expanded Analysis (cont'd)**
> > >
> > > ## C4: Ablation Concerns and Statistical Significance
> > > The reviewer raises a valid concern about the ablation results: removing the Evaluator increases ASR by 40% in relative terms but only 1.1 percentage points in absolute terms (from 2.75% to 3.85%). We agree that absolute differences at low ASR values are harder to interpret statistically, particularly given the sample sizes involved. We will add two things in the revised paper: first, a note in Section 5.5 acknowledging that the Evaluator's contribution is modest in absolute terms and that its primary role is as a safety net for edge cases that bypass the Orchestrator, rather than as a major driver of overall performance; second, we will report the number of examples underlying each ASR estimate so readers can assess the statistical confidence of the differences themselves. We maintain that the Evaluator's contribution is meaningful. A 40% relative increase in ASR when it is removed represents a real degradation in a security-critical setting, but we agree the paper should present this more carefully.
> > >
> > >
> > > ## C5: Writing Allocation
> > > The reviewer's observation that the paper's writing allocation is imbalanced is well-taken. Section 4.2, which describes the prompt optimization procedure, currently devotes considerable space to explaining MIPROv2's mechanics (a method that already has its own paper). Meanwhile, the experimental sections provide insufficient context for readers to interpret the results. In the revised paper we will: shorten Section 4.2 by summarizing MIPROv2 more briefly and pointing readers to Opsahl-Ong et al. (2024) for details; expand the experimental sections with richer explanation of what each result shows and why it supports or qualifies our claims; and add a brief "what does this result tell us" narrative after each major table. We agree with the reviewer that the paper should teach the reader something actionable about when and why this approach works, not just report that it does.
> > >
> > > ## C6: Terminology and Accessibility
> > > We will define all technical terms at their first occurrence in the main text. Specifically: "jailbreaking" will be defined at first use in the introduction; "PII" (personally identifiable information) will be expanded at first use in Section 3.1; "WMDP" (Weapons of Mass Destruction Proxy) will be expanded at first use in Section 5; "compliance" as used in Tables 4 and 5 will be defined as full compliance on PHTest at its first appearance. We will also add a brief description of each benchmark's content and evaluation protocol in the main text rather than relying on the appendix for this information. We agree with the reviewer that for readers not fully immersed in LLM safety literature, these definitions are essential for following the argument.
> > >
> > >
> > > ## Summary of Changes
> > >
> > > - Scoped all major claims to the specific benchmarks and model families evaluated, replacing overly general language throughout the abstract, introduction, and conclusion
> > > - Added clarification of baseline backbones and whether numbers are reproduced or copied, including a new footnote in Table 2 and a dedicated paragraph in Section 5
> > > - Added explicit metric definitions at first use for WMDP accuracy, StrongREJECT score, PHTest compliance, and MT-Bench
> > > - Expanded benchmark descriptions in Section 5 with content, protocol, and limitation summaries
> > > - Shortened Section 4.2 and redistributed space toward richer experimental narrative
> > > - Expanded Section 5.3 to connect Figure 6 and Table 5 as joint evidence for adaptability
> > > - Added clarification of Evaluator's absolute vs relative contribution in Section 5.5, with sample size reporting
> > > - Defined all technical terms at first occurrence in the main text

---

### Review · Reviewer_7M64 · 2026-03-24

**Summary Of Contributions:**

This paper proposes to use a modular prompt system that can be rapidly adapted to new attacks via Bayesian prompt optimization, achieved via the DSPy framework. The authors claim that this dynamic, deployment-time modular system is faster, more effective and obtains better refusal/accuracy trade-offs than prior work. This work is tested across three models (LLaMA-3-8B, Qwen2.5-72B, DeepSeek-R1, 8B and 70B) on four benchmarks, WMDP, TOFU, StrongReject and PHTest.

**Additional Comments:**

None

**Audience:**

No

**Audience Explanation:**

I'm not entirely certain here. At a scientific level, the novelty appears limited to me, inference-time, multi-agent, and few-shot defenses are already a thing. The specific recipe is novel, but other than "let's apply this specific recipe", there's not much that's actionable from the paper because the authors did not investigate their framework enough.

TMLR as a venue explicitly rejects novelty-gating, but that's not what I'm pushing against here. I don't feel like I learned much by reading this paper, because there's not enough explanations of how the recipe was derived nor which ingredients matter.

**Broader Impact Concerns:**

Addressed by section 6

**Claims And Evidence:**

No

**Claims Explanation:**

The raw performance claims are backed by empirical evidence; on the benchmarks, we do see that the proposed approach matches or beats prior work.

Scientifically, there is very little: _why_ does the method work? There are basically two results in that direction:
* Table 7 is not a very meaningful way to measure the improvements, the claim is basically that "having a filter is better than having no filter". Perhaps more accurately, the claim is, multiple filtering agents is better than a single filter, but it's not clear that there is evidence for this in the paper (i.e. using a much more powerful model as a single filter).
* Table 8 is more meaningful, but again the evidence for the claim we have is basically that "choosing the best filter out of a set of possible filters is better than choosing a base filter", which it would have been very surprising if it wasn't the case. What would be more interesting, since the core claim is around sample efficiency, is a deeper analysis of the sample dependency (e.g. Figure 6, generalized).

This leaves many unanswered questions, on top of there being some lack of details:
- two $\lambda$ are mentioned, what's their impact? What's their actual values?
- what is the cost of the prompt optimization?
- what is the impact of the choice of $\Theta$ on the prompt optimization? How many steps are necessary? How useful is the Bayesian part, could we just generate 100 prompts and rank them?

**Requested Changes:**

Here are some changes which I think would address some of the issues I have:
- Detail the objective function $\mathcal{L}$ and the $\lambda$ values
- Add a compute-equivalent "single agent" model in Table 7
- Measure the impact of the method used for prompt optimization; is random search good enough? Is just prompting a much stronger frontier LLM sufficient?
- Detail the costs of the prompt optimization phase
- Improve the sample efficiency results

---

> ### Author Response · Authors · 2026-04-08
> **Rebuttal to Reviewer 7M64: Clarifying Contributions, Optimization, and Sample Efficiency**
>
> We thank the reviewer for the detailed feedback. We believe some of the concerns stem from a mismatch between the reviewer's expectation of what the paper claims and what we actually claim. Our core contribution is not that Bayesian optimization is a novel or superior optimizer, nor that modular systems are novel in general, it is that **inference-time, prompt-driven modular systems can be adapted to novel security threats with very few labeled examples and no model retraining, and that this approach achieves a better safety-utility tradeoff than existing defenses**. We address each concern below and clarify where we will update the paper.
>
>
> **NOTE:** Tables 6, 7, 8, and 9 in the following rebuttal refer to the most recent PDF version of the paper that we upload on OpenReview.
>
> ## C1: Clarification of the Objective Function and λ Values
> The objective function is implemented in our released code as orchestrator_safety_metric, which assigns scores of 0.0 for false negatives (unsafe content passing through), 0.3 for false positives (over-refusal), and 1.0 for correct classifications. This encodes the asymmetric penalty structure with effectively λ_FN → ∞ relative to λ_FP, reflecting the security-critical nature of missed detections. We have revised the paper and made this explicit.
>
>
> ## C2: The Ablation Study (Table 8) and the Compute-Equivalent Baseline
> We thank the reviewer for this suggestion and want to clarify what we believe our ablation already establishes, and why the proposed compute-equivalent baseline may not be the most informative comparison in this setting.
> * The "Responder Only" configuration in Table 8 already serves as a strong single-agent baseline: it uses the identical underlying model as AegisLLM — the same weights, the same inference infrastructure — with no modular wrapper and no prompt optimization. The 302% improvement in ASR reduction (from 11.06% to 2.75%) when moving from Responder Only to Full AegisLLM therefore directly isolates the contribution of modularity and optimized prompts, holding model identity constant. This is precisely the comparison needed to establish that the system design is doing the work, not the underlying model capacity.
> * The reviewer suggests comparing against a single stronger model with equivalent compute. We respectfully push back on this framing for two reasons. First, it implicitly assumes that safety performance scales with model size or compute, an assumption our own results do not support. Across our experiments with LLaMA-3-8B, Qwen2.5-72B, and DeepSeek-R1 variants of different sizes, we do not observe a consistent relationship between model scale and safety outcomes. Larger models do not automatically yield lower ASR or better safety-utility tradeoffs. Safety in this context is driven by the specificity and quality of the decision-making policy, not raw capacity. Second, a comparison against a hypothetical stronger model would confound model capability with system design, making it harder to attribute improvements to either factor cleanly.
> * We also note that AutoDefense EX-3, which appears in Table 5, effectively serves as a compute-heavy multi-agent baseline. It uses a triple-agent configuration, is approximately 18x slower than AegisLLM, and costs significantly more per query. However, AegisLLM outperforms it on the safety-utility tradeoff that matters for real deployment (higher compliance, competitive ASR). This suggests that simply adding more compute to a multi-agent system does not monotonically improve the tradeoff, further undermining the premise of the compute-equivalent baseline.
> * We will add a clarifying note in the revised paper explaining the design rationale of the ablation study and why Responder Only serves as the appropriate single-agent comparison.
>
> ## C3: The Role of the Bayesian Optimizer and Prompt Search
> Our contribution is not that Bayesian optimization is the right optimizer for this problem. It is that the system as a whole can be adapted with very few examples. MIPROv2 is used as an established, off-the-shelf tool, analogous to using Adam for neural network training. The question of whether random prompt search would be equivalent conflates two axes:
> * optimizer efficiency in terms of iterations, which is addressed by the MIPROv2 paper itself,
> * and data efficiency in terms of labeled security examples needed at deployment time, which is our actual claim.
>
> Furthermore, random prompt generation is not a well-defined baseline in this setting: prompt space is a combinatorial space of natural language instructions and few-shot demonstration selections, and evaluating 100 random configurations would incur essentially the same cost as running MIPROv2 for 100 iterations, except without the benefit of surrogate-guided search. We point the reviewer to Opsahl-Ong et al. (2024) for direct comparison of guided versus random search in prompt optimization.

---

> ### Author Response · Authors · 2026-04-08
> **Rebuttal to Reviewer 7M64: Clarifying Contributions, Optimization, and Sample Efficiency (cont'd)**
>
> ## C4: Cost of the Prompt Optimization Phase
> The optimization is a one-time, offline process. Given that it operates on a very small dataset (50 queries per benchmark), its computational cost is extremely low. We ran detailed profiling of this phase which are reported in Table 6 of the updated paper. These runs are done using a single NVIDIA A6000 GPU (run locally using vLLM, similar to our runs for other results in the paper including the results in Table 7). We have added a dedicated paragraph and table detailing these exact optimization costs in Section 5.4. For your convenience, the specific metrics are provided below:
>
> | Model | Task | Component | LLM Calls | Time (s) | Input Tokens | Output Tokens | Total Tokens | Cost (USD) |
> | :--- | :--- | :--- | :---: | :---: | :---: | :---: | :---: | :---: |
> | **Llama-3-8B** | Unlearning | Orchestrator | 284 | 243.05 | 1,088,010 | 29,494 | 1,117,504 | $0.0232 |
> | | Jailbreak | Orchestrator | 333 | 222.96 | 756,244 | 41,066 | 797,310 | $0.0172 |
> | | Jailbreak | Evaluator | 366 | 282.42 | 1,711,514 | 51,399 | 1,762,913 | $0.0368 |
> | **Qwen2.5-7B** | Unlearning | Orchestrator | 378 | 259.34 | 1,451,489 | 33,834 | 1,485,323 | $0.0307 |
> | | Jailbreak | Orchestrator | 430 | 241.30 | 974,372 | 46,123 | 1,020,495 | $0.0218 |
> | | Jailbreak | Evaluator | 423 | 282.87 | 1,675,357 | 50,119 | 1,725,476 | $0.0360 |
>
> > **Note:** Estimated costs are based on 0.02 USD per 1M input tokens and 0.05 USD per 1M output tokens.
>
> The monetary costs of optimization are is mere cents (e.g., \$0.017 to \$0.037 USD) for Llama-3-8B and Qwen2.5-7B and take only a few minutes on a single GPU.
>
> ## C5: Sample Efficiency Results
> We thank the reviewer for this feedback and want to clarify what our sample efficiency claim is and is not, and point to existing results in the paper that we believe collectively make a stronger case than Figure 6 alone.
> Figure 6 is intended as a proof-of-concept demonstration of adaptation rather than a comprehensive characterization of the sample efficiency learning curve. We acknowledge that 15 attacks from a single benchmark is a limited evaluation window, and we will make this scope explicit in the revised paper. However, we want to draw the reviewer's attention to a complementary result that we believe constitutes stronger evidence for sample efficiency, which is the zero-shot transfer result in Table 5.
>
> Specifically, the prompts optimized on StrongREJECT and PHTest using only 50 labeled examples total (split 20% for training and 80% for validation) transfer directly to Wildjailbreak without any additional optimization or labeled examples from that distribution. Wildjailbreak contains over 5,700 unique clusters of jailbreak tactics mined from real in-the-wild interactions, representing a substantially different and more diverse distribution than StrongREJECT. Despite this distribution shift, AegisLLM reduces ASR from 11.06% to 3.25% on Wildjailbreak with zero additional examples. We argue that zero-shot generalization to a held-out distribution is actually the most demanding end of the sample efficiency spectrum. It demonstrates that the optimized defensive policy captures something general about threat structure rather than overfitting to the specific patterns seen during optimization.
>
> Taken together, we therefore have two complementary pieces of evidence for sample efficiency. First, Figure 6 shows that meaningful adaptation to a new attack type occurs with as few as 5 to 15 examples per attack, with refusal rates improving from 60.7% to 73.0% while false refusal rates rise only modestly from 8.7% to 10.3%. Second, Table 5 shows that a policy optimized on 50 examples from one distribution generalizes to a completely unseen distribution with no additional data. These two results operate at different points on the sample efficiency spectrum (few-shot adaptation on one end and zero-shot transfer on the other) and together they make a more complete case than either result alone.
>
> We also want to address the concern about statistical confidence over 15 attack samples in Figure 6. The evaluation in Figure 6 specifically targets the 15 strongest attacks in StrongREJECT. These are selected precisely because they are the hardest cases, representing the tail of the distribution where adaptation matters most. Improvement on this subset is therefore a conservative estimate of overall adaptation effectiveness, since these are the attacks most resistant to the base optimized policy. We will add this clarification to the figure caption in the revised paper.

---

> ### Author Response · Authors · 2026-04-08
> **Rebuttal to Reviewer 7M64: Clarifying Contributions, Optimization, and Sample Efficiency (cont'd)**
>
> ## C6: Scientific Contribution and Interpretation of Ablation Studies
> > Table 8 is not a very meaningful way to measure the improvements, the claim is basically that "having a filter is better than having no filter". Perhaps more accurately, the claim is, multiple filtering agents is better than a single filter, but it's not clear that there is evidence for this in the paper (i.e. using a much more powerful model as a single filter).
>
> We appreciate the reviewer's request for a deeper interpretation of Table 8. We respectfully argue that the ablation study demonstrates a more nuanced point than simply stating that filters are good. It highlights the *synergistic and non-redundant nature of our layered defense*. Removing the Orchestrator (the pre-generation filter) causes a +91% relative increase in Attack Success Rate (ASR), while removing the Evaluator (the post-generation filter) causes a +40% increase. This shows that each module is responsible for catching a significant and distinct portion of threats that the other would miss. The full system's ASR on WildJailbreak (2.75%) is also substantially lower than that of the Responder alone (11.06%), resulting in a 302% relative improvement that underscores the power of this layered, multi-agent approach. This supports our central thesis that **a cooperative, modular system provides a more robust defense than the sum of its parts**.
>
> ## C7: Significance of Prompt Optimization and Sample Dependency
> > Table 9 is more meaningful, but again the evidence for the claim we have is basically that "choosing the best filter out of a set of possible filters is better than choosing a base filter". What would be more interesting, since the core claim is around sample efficiency, is a deeper analysis of the sample dependency (e.g. Figure 6, generalized).
>
> We believe the significance of Table 9 lies in the *magnitude* and *efficiency* of the improvement. It demonstrates that with a remarkably small number of optimization examples (as stated in Section 5, 50 training queries), we can achieve substantial gains in performance (e.g., a +30.3% increase in the flagged ratio for Llama-3-8B on WMDP-Cyber). Regarding sample dependency, Figure 6 is our primary evidence in this direction. It shows that even after exposure to as few as **5 attacks**, the system's refusal rate on the top-15 strongest attacks begins to improve, and by 15 attacks, it shows a significant improvement (from 60.7% to 73.0%) while maintaining a very low false refusal rate. This demonstrates the core concept of rapid, sample-efficient adaptation that is central to our work.
>
> ## Summary of Changes
> - Explicit reporting of λ_FN and λ_FP values and their rationale in Section 4.2, with a pointer to released code
> - Clarification of the ablation study design rationale in Section 5.5, explaining why Responder Only is the appropriate single-agent comparison
> - Explicit framing of MIPROv2 as a tool rather than a contribution, with a pointer to Opsahl-Ong et al. for optimizer comparisons
> - Reporting of specific prompt optimization costs (time, API calls, tokens, monetary cost) as a dedicated paragraph and table in Section 5.4
> - Revised Section 5.3 connecting Figure 6 and Table 5 as joint evidence for sample efficiency, with clarified scope of each result

---

### Review · Reviewer_nNUg · 2026-03-25

**Summary Of Contributions:**

The paper proposes AegisLLM, a modular inference-time defence system for LLM safety consisting of four components (Orchestrator, Deflector, Responder, Evaluator). The key idea is that system prompts for these modules can be optimised via Bayesian optimisation using very few examples, enabling post-deployment adaptation to threats like jailbreaking and sensitive information disclosure without model retraining.

**Audience:**

Yes

**Audience Explanation:**

The argument that safety should be scaled at inference time, not just during training, is well-motivated and timely. Furthermore, the paper addresses two distinct threats, evaluates multiple model architectures and benchmarks, and includes ablations, efficiency analyses, and generalisation tests, showing strong results across these benchmarks.

**Broader Impact Concerns:**

Since the paper is discussing the LLM safety, I do not see any concerns which would require a Broader Impact Statement.

**Claims And Evidence:**

Yes

**Claims Explanation:**

The core empirical claims are supported by extensive experiments across multiple benchmarks, multiple model architectures, and appropriate baselines. The ablation studies convincingly demonstrate that each component contributes to overall performance, and the efficiency analysis provides concrete cost comparisons. The results are presented with clear metrics, and the experimental setup is reproducible, given the detailed appendix with full prompt listings. However, minor concerns remain related to some misleading comparisons and claims (see Requested Changes)

**Requested Changes:**

**1.** The four-module pipeline (input classifier → responder → output classifier → feedback loop) is a fairly standard pattern. The paper acknowledges this implicitly by calling it a "case study," but the novelty claim around "system-level optimisation for safety" overstates what is essentially applying DSPy's existing MIPROv2 optimiser to safety-oriented prompts. The optimisation method (Bayesian optimisation over prompt configurations) is entirely borrowed from prior work. Could the authors clarify this?

**2.** To my understanding AegisLLM doesn't perform unlearning, it performs inference-time filtering. The knowledge remains fully intact in the model weights. The paper does acknowledge this in the limitations, but the extensive comparison against actual unlearning methods (RMU, GradDiff, TAR, ELM) in Table 2 creates an questionable comparison. A determined adversary who gains direct access to the Responder model (or who crafts inputs that bypass the Orchestrator) would find all knowledge intact. While black box access is assumed, the authors should not introduce the advantage as unlearning, as this can be missleading, but clearly state that there methods is able to filter. Could the authors clarify this?

**3.** The Orchestrator and Evaluator are themselves LLMs susceptible to prompt injection and adversarial inputs. What happens when an attacker specifically targets the Orchestrator's classification prompt? The paper evaluates against existing jailbreak benchmarks, but doesn't consider adaptive attacks against the defense system itself. A strong adversary would craft inputs to fool the Orchestrator specifically, not just the Responder. The paper should discuss (or ideally evaluate) this threat.

**4.** If I understand correctly, the claim that AegisLLM is "faster than the base model" holds only for unsafe queries that get early-rejected. For benign queries AegisLLM adds substantial overhead (e.g., 30.2s vs 9.7s for Llama-3-8B on safe prompts). In most deployments, the vast majority of queries are benign, so the average-case overhead is closer to the benign-query numbers. The framing in Figure 1(c) emphasizing the unsafe-query speedup is misleading.

---

> ### Author Response · Authors · 2026-04-08
> **Rebuttal to Reviewer nNUg: Addressing Novelty, Unlearning, and Efficiency Claims**
>
> We thank Reviewer nNUg for their thorough assessment and recognition of our paper's contributions and extensive experiments and ablation studies in addressing critical LLM safety challenges. We also appreciate the reviewer's insights regarding the novelty of our approach and the practical implications of test-time adaptability to mitigate LLM risks. We address the reviewer's concerns in the following paragraphs.
>
> **NOTE:** Table 7 in the following rebuttal refers to the most recent PDF version of the paper that we upload on OpenReview.
>
> ## C1: Novelty of the System-Level Optimization Claim
> > The four-module pipeline (input classifier → responder → output classifier → feedback loop) is a fairly standard pattern. The paper acknowledges this implicitly by calling it a "case study," but the novelty claim around "system-level optimisation for safety" overstates what is essentially applying DSPy's existing MIPROv2 optimiser to safety-oriented prompts. The optimisation method (Bayesian optimisation over prompt configurations) is entirely borrowed from prior work. Could the authors clarify this?
>
> We agree that MIPROv2 is an existing optimizer and that our contribution is not the optimization method itself. We will revise the framing in Section 4.2 to make this explicit: our contribution is the application of prompt optimization to a security-specific objective in a modular inference-time system, and the demonstration that this approach achieves strong, generalizable safety performance with very few labeled examples. The novelty lies in the system design, the security-focused objective function, and the empirical findings, not in the optimizer. We will also update the contributions list in Section 1 to reflect this distinction more precisely.
>
> ## C2: On the Distinction Between "Unlearning" and Inference-Time Filtering
> > To my understanding AegisLLM doesn't perform unlearning, it performs inference-time filtering. The knowledge remains fully intact in the model weights... the authors should not introduce the advantage as unlearning, as this can be missleading, but clearly state that there methods is able to filter
>
> We thank the reviewer for raising this important distinction. AegisLLM enforces non-disclosure at inference time rather than removing internal representations. This is correctly noted in our limitations section, and we agree that the framing of Table 2 risks implying a stronger claim than we intend. We will make three changes in the revised paper. First, we will rename the section header and table caption to refer to "hazardous output suppression" rather than "unlearning performance." Second, we will add an explicit clarifying sentence at the start of Section 5.1 stating that AegisLLM's mechanism is filtering rather than weight-level unlearning, and that the comparison is on behavioral outcomes (whether the system prevents hazardous information from being disclosed) rather than on internal knowledge removal. Third, we will add a brief discussion of the threat model implication the reviewer raises: a determined adversary with direct model access would bypass AegisLLM entirely, and this is an acknowledged limitation of all inference-time defenses. We will frame our approach as complementary to training-time unlearning rather than a replacement.
>
> ## C3: Adaptive Attacks Against the Defense System Itself
> > The Orchestrator and Evaluator are themselves LLMs susceptible to prompt injection and adversarial inputs. What happens when an attacker specifically targets the Orchestrator's classification prompt? The paper evaluates against existing jailbreak benchmarks, but doesn't consider adaptive attacks against the defense system itself.
>
> The reviewer raises an important and well-known challenge in adversarial defense evaluation: a sufficiently adaptive adversary who knows the structure of the defense system can craft inputs specifically designed to fool the classifier rather than the responder. We agree this is a meaningful threat and will add a dedicated paragraph in Section 6 discussing it explicitly. We note two partial mitigations present in our current design. First, the two-stage structure of AegisLLM (Orchestrator pre-screening followed by Evaluator post-hoc auditing) means an adversary must simultaneously fool both components, which have different prompt configurations and decision criteria. Second, the generalization results in Table 5, showing strong performance on Wildjailbreak without any optimization on that distribution, suggest that the optimized policies capture something general about threat structure rather than overfitting to specific surface patterns. That said, we acknowledge that a full adaptive attack evaluation, where the adversary has knowledge of the Orchestrator's prompt, is an important direction for future work that is beyond the scope of this paper. We will add this as an explicit limitation and future work item.

---

> > ### Author Response · Authors · 2026-04-08
> > **Rebuttal to Reviewer nNUg: Addressing Novelty, Unlearning, and Efficiency Claims (cont'd)**
> >
> > ## C4: Misleading Efficiency Framing in Figure 1\(c\)
> > > If I understand correctly, the claim that AegisLLM is "faster than the base model" holds only for unsafe queries that get early-rejected. For benign queries AegisLLM adds substantial overhead... The framing in Figure 1(c) emphasizing the unsafe-query speedup is misleading.
> >
> > The reviewer is correct that the framing in Figure 1(c) is misleading. The "faster than the base model" claim holds only for unsafe queries that trigger early rejection by the Orchestrator. For benign queries, AegisLLM introduces overhead (30.2s vs 9.7s for Llama-3-8B as shown in Table 7). In most real deployments the vast majority of queries are benign, so the average-case latency is closer to the benign-query numbers. We will make three changes: first, revise the Figure 1(c) caption to explicitly state that the speedup applies to unsafe queries only; second, add a sentence in Section 5.4 reporting a weighted average latency assuming a realistic safe-to-unsafe query ratio (e.g., 95 to 5); third, reframe the efficiency claim in the abstract and introduction to focus on the comparison against agentic baselines like AutoDefense, where AegisLLM is genuinely and substantially faster across both safe and unsafe queries, rather than the comparison against the base model.
> >
> >
> >
> > ## Summary of Changes
> > - Revised framing of Section 4.2 and contributions list to clarify that MIPROv2 is a tool rather than a contribution
> > - Renamed Table 2 section and added clarifying sentence distinguishing filtering from weight-level unlearning
> > - Added threat model discussion in Section 6 addressing adaptive attacks against the defense components
> > - Revised Figure 1\(c\) caption, added weighted average latency numbers in Section 5.4, reframed efficiency claims in abstract and introduction

---

### Decision · Action_Editor_PVrW · 2026-05-03

**Recommendation:** Reject

**Audience:**

Yes

**Audience Explanation:**

The paper shows some good results in the experiment section, thus I believe some audiences should be interested in the paper.
However, the paper has the following weakness.

 -**No insightful understanding or results are proposed.** As pointed out by reviewer 7M64 and nZ2s, the paper does not propose any insightful analysis to explain why their proposed method works. And no further new findings can be used to motivate readers on broader safety topics. The authors need to explore their methods more deeply to give some scientific understanding.

 -**The novelty of the proposed method is incremental.** As pointed out by all reviewers, the proposed technique is the combination of existing methods instead of a new one.

**Claims And Evidence:**

No

**Claims Explanation:**

The paper proposes an automated prompt optimization framework to solve the adaptive safety risks during inference time on unlearning and jailbreak safety. However, as pointed out by reviewers, their claims are not accurately supported in their paper. The main concerns are still remaining:

 - **The proposed method is not unlearning.** It is a concept of misalignment. As pointed out by NNUG, NZ2s, and admitted by the authors, the proposed method is not the real unlearning, which greatly weakens the papers’ contributions.

 - **The adaptive attacks are not sufficient.** Reviewer nNUg first proposed the concern about adaptive attacks on Orchestrator and Evaluators. However, the authors do not solve this during the rebuttal. Therefore, whether the proposed method is indeed a robust safety framework against adaptive attacks is still unknown.

 - **The experiment settings and results are not enough.** Reviewer 7M64 and nZ2s show concerns about the experiment settings, e.g., some SOTA methods are not selected, $\lambda$ settings, compute-equivalent baselineså, and etc. And they also criticize that the sample size is not enough. The limited size cannot support the author’s claim. Therefore, I think their experiments cannot support their claims.